# Marine Gelatine from Rest Raw Materials

**Ivan Milovanovic and Maria Hayes ***

Teagasc, The Irish Agricultural and Food Development Authority, Food BioSciences Department, Ashtown, Dublin 15, Ireland; ivan.milovanovic@teagasc.ie
* Correspondence: maria.hayes@teagasc.ie; Tel.: +353-1-805-9957

**Abstract:** In recent years, demand for consumption of marine foods, and especially fish, has substantially increased worldwide. The majority of collagen available is sourced from mammalian-derived products. Although fish derived gelatine is a viable alternative to mammalian sourced gelatine, there are certain limitations related to the use of fish gelatine that include odour, colour, functional properties, and consistency in its amino acid composition. Chemicals used for pre-treatment, as well as extraction conditions such as temperature and time, can influence the length of polypeptide chains that result and the functional properties of the gelatine. Compared to traditional sources, gelatines derived from fish show significant differences in chemical and physical properties, and great care should be paid to optimization of the production process in order to obtain a product with the best properties for intended applications. The focus of this review is to explore the feasibility of producing gelatine sourced from marine processing by-products using different pre-treatment and extraction strategies with the aim of improving the techno-functional properties of the final product and improving the clean-label status of gelatines. The bioactivities of gelatine hydrolysates are also discussed.

**Keywords:** gelatine; marine; by-products; fish; industry; extraction

---

## 1. Introduction

The consumption of marine foods, and especially fish, has seen a significant increase in demand worldwide during the recent decades. This increase can be mainly attributed to the recognition of fish as important in human health [1]. Another important factor is globalization of world food trade, which has resulted in lower prices and better accessibility of marine commodities around the world. Fish consumption worldwide has seen an annual increase at an average rate of 3.2% since the early 1960s [2], and this trend is likely to follow the growing global demand, driven by the increase in human population and consumer purchasing power. Production of gelatine is becoming an increasingly interesting perspective of adding economic value to by-products generated by the fishing industry.

The majority of collagen available is sourced from mammalian-derived products including pig skin, cattle hide, and cattle bones. Hayatudin [3] reports that approximately 41% of the gelatine produced in the world is sourced from pig skin, 28.5% from bovine hides, and 29.5% from bovine bones. The production of fish-derived gelatine currently accounts for only 1.5% of total annual gelatine production worldwide, which is estimated to be around 270,000 metric tonnes [4].

The European Union has introduced the Common Fisheries Policy (CFP). This current policy stipulates that between 2015 and 2020 catch limits should be set that are sustainable and which can maintain fish stocks in the long term. The CFP has four principle policy areas: (1) fisheries management, (2) international policy (3), market and trade policy, and (4) funding policy. An important part of the fisheries policy is related to the discards and landing obligation. Discarding is the practice of returning unwanted catches to the sea (either dead or alive), due to lack of market demand, undersized

fish samples, or because of the catch composition rules. The aim of the CFP is to first gradually and then completely eliminate the practice of wasteful discarding. This should be attained through the implementation of the landings obligation for all common fisheries from 2015 to 2019. The landing obligation requires all catches of regulated commercial species on-board to be landed and counted against quota, with undersized fish specimens that cannot be marketed for direct human consumption, and obligation of certain protected species to be returned back to the sea. By 2019 all species subject to TAC (Total Allowance Catch) limits and Minimum Conservation Reference Sizes in the Mediterranean will be subject to the landing obligation [5].

*Opportunities for by-Catch Utilization*

By-products from marine processing industry represent a major environmental and economic challenge due to inadequate disposal options and/or costs associated with disposal at landfills. Processing leftovers such as bloodwaters, trimmings, fins, frames, heads, shells, skin, viscera, and stickwater/effluent are currently used in Ireland for the production of fish meal, fish oil, fertilizer, and animal feeds [6]. Solid waste from surimi processing, which can amount for 50 to 70% of the utilized raw material, is also an important source of by-products [7]. Boarfish (*Capros aper*) and blue whiting (*Micromesistius poutassou*) are two pelagic species which represent specific challenges for the fish processing industry. They are currently viewed as lower value species, due to their small size which makes their processing demanding, although some advances have been made in the field of production of blue whiting skinless fillets [8]. Another option for processing of these species would be for production of surimi products, especially in the case of small specimens which are unsuitable for machine filleting operations.

Boarfish (Figure 1) [9] is a small species (up to 23 cm in length) of mesopelagic shoaling fish, characterized by its orange colour, large eyes, and protrusible mouth. They can be found inhabiting shallow seas and shelf slopes from 400–600 m depth, and are widely distributed across the eastern Atlantic from Norway to Senegal, including the Mediterranean region [10]. Although it is considered a sub-tropical fish species, in recent decades boarfish has become very abundant throughout its range, which may be explained by rising ocean temperatures due to climate change [11]. Although the 2017 boarfish quota for Ireland is 36% lower than previous year's quota, the allowed 18,850 tonnes limit is still among the highest among European countries [12]. The main utilization of landed boarfish in Ireland includes export to Denmark for production of fishmeal [13], but other potential uses are also considered. The Irish Sea Fisheries Board (Bord Iascaigh Mara, BIM) currently recommends use of Boarfish for direct human consumption, with marketing options either in the form of commodity products including 20 kg blast frozen blocks of mince or as a headed and gutted product suitable for frying [14]. Other authors have recently discussed alternative means of boarfish biomass exploitation, which include hydrolysis of its proteins to obtain protein hydrolysates and extraction of valuable peptides and biomolecules [15,16]. However, large-scale production of gelatine from boarfish by-products is not sufficiently researched as an option for valorisation of this biomass currently.

The focus of this review is to explore the feasibility of producing gelatine sourced from marine processing by-products specifically from blue whiting and boarfish processing by-products including skins and bones.

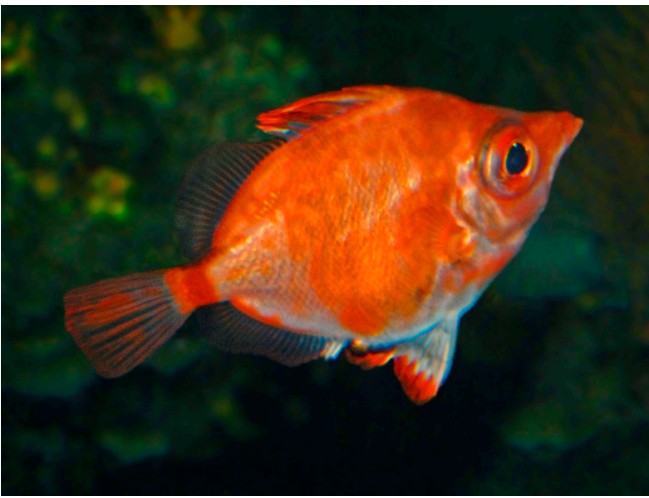

**Figure 1.** Boarfish (*Capros aper*) [9].

## 2. Properties and Applications of Marine-Derived Gelatine

Gelatine is a water-soluble protein produced by partial hydrolysis of collagen, a ubiquitous constituent of bones, cartilage tissues, and skins of animals [7]. Collagen is the principal protein constituent found in skin, cartilage, blood vessel walls, and teeth. Together with other proteins, such as elastin and proteoglycans, it builds extracellular matrix around cells in various tissues. The collagen molecule is shaped in form of a triple helix, formed by three α-chains positioned in a three-dimensional structure which allows efficient intramolecular hydrogen bonding [6]. Although all of the 20 natural amino acids can be found in collagen, it is particularly rich in glycine, proline, and hydroxyproline. Tri-peptides which build the main portion of collagen have frequent repetitions of the sequence—Gly-Pro-X or Gly-X-Hyp and the distribution of polar and non-polar amino acid residues at the X position determines the order of aggregation of the molecule. Polypeptide chains undergo a total or partial separation during denaturation of collagen, due to weakening and destruction of hydrogen bonds, which causes a loss of the original triple-helix conformation. This results in polymers adopting a coiled form different than the original protein structure [17]. The proportion and ratio of total amino acids influences the gelatine stability, and this property may vary depending on the source of collagen. The current manufacturing process of gelatine includes pre-treatment of raw animal materials (hides, bones, etc.) with dilute acid or alkali, which results in partial denaturation of collagen due to hydrogen bonds cleavage in its structure [18]. It is known that various marine processing by-products, such as fish skin, bones, scales, surimi production discharge waste, and squid skin represent a good source of collagen for gelatin production [17,19–21].

In general, gelatine is used in the food, pharmaceutical, and photography industry for a number of applications including jelly production, encapsulation, and fruit juice clarification, dairy processing, soup manufacture, photography and others. Typical applications of gelatine depend on the gelatine type, and some are shown in Table 1 [22]. Its great versatility enables use in both the food and pharmaceutical industry. In the food industry, gelatine is considered an essential ingredient, and can also be considered a "clean label" product, since

- Gelatine is not chemically modified and has no, possibly harmful, by-products of chemical modification
- It does not contain and is not made of any genetically modified organisms
- Is not a food additive and therefore does not require an E-number
- It is considered Generally Recognised As Safe (GRAS)
- It does not cause any known allergies
- It has been consumed for more than 2000 years and is known for generations [23]

**Table 1.** Usage of gelatine depending on type [22].

| Type of Gelatine | Typical Usage |
| --- | --- |
| Food grade | Confectionary, gelatine desserts, gelatine in meats, clarification of beverages and juices, special dietary uses |
| Pharmaceutical | Gelatine capsules (hard and soft type), tablets and tablet coating, suppositories, gelatine emulsions, microencapsulation, absorbable gelatine sponge and films, plasma substitute, pastilles and troches, bacterial growth media |
| Photographic | Photographic emulsions |
| Other (technical) | Coating and sizing, paper manufacture, printing processes, colloidal applications, matches, coated abrasives, adhesives, films and light filters, cosmetics, microencapsulation |

## 2.1. Legislative and Safety Considerations of Marine Gelatine Production

Although production and use of gelatine is a highly regulated field, additional challenges may lie ahead due to changes in consumer trends in recent times. Edible gelatine must meet the requirements laid by the Food Hygiene Regulation (EC) No 853/2004 (also Commission Regulation (EU) 2016/355 of 11 March 2016 amending Annex III to Regulation (EC) No 853/2004) and is additionally subject to European food regulations [23]. Pharmaceutical gelatine, in addition to these requirements, must also fulfill the conditions laid down by specific pharmacopeias. The Regulation (EC) No 853/2004 prescribes the necessary critical points of control during gelatine manufacture. It addresses all aspects, from the raw materials to the delivery of the final product: origin, transport, and storage of raw materials; manufacturing conditions; chemical requirements for gelatine and collagen peptides; as well as packaging, storage, and transport. The important safety parameters, such as levels of heavy metals and toxic contaminants and microbiological safety are covered by this regulation and complementary regulations, such as (EC) No. 2073/2005 [24]. Gelatine used for pharmaceutical purposes must comply with specific additional regulations. Gelatine is a well-known material with an excellent safety record and is GRAS for human use [25]. Other chemicals typically used for gelatine production are known and approved food additives which do not possess chronic toxicity and include hydrochloric acid (E507), citric acid (E330), sodium hydroxide (E524), and calcium hydroxide (E526). Some of the enzymes which can be used for gelatine production, such as proteases from *Aspergillus oryzae*, are also included in the list of approved food additives under Regulation (EC) No 234/2011 [24]. Additionally, since the process of gelatine manufacture includes washing of the material after every treatment step, as well as purification of the gelatine solution itself, these chemicals and enzymes are removed from the final product.

Fish and fish products are known to be a common source of allergic reactions in consumers. The Regulation (EU) No 1169/2011 on the provision of food information to consumers has entered into application on 13 December 2014. Under this regulation, the obligation to provide nutrition information, as well as stating the possible food allergens is mandated. Fish and fish products (except fish gelatine used as a carrier for vitamin or carotenoid preparations and fish gelatine or Isinglass used as fining agent in beer and wine) must be declared if present in the food. Fish allergy is a pathophysiological immune response to specific fish proteins, mediated by IgE-type antibodies. Humans can become sensitized by allergen exposure via the gastro-intestinal tract during ingestion, which is the major route of sensitization, or via the respiratory system by fish aeroallergens or skin contact [26]. Parvalbumins are recognized as the most important group of fish proteins with allergic potential, but other proteins, such as collagen, transferrin, fish enolases, and aldolases have also shown allergic potential. Parvalbumins are highly stable, low-molecular-weight proteins (10–12 kDa), which are mostly found in fish muscle, but their content is significantly lower in pelagic fish compared to warm water and freshwater species, since the highest concentrations can be found in white muscle tissue [26]. Also, during the recent years, 50 kDa enolases and 40 kDa aldolases were identified as important fish allergens in cod, salmon, and tuna [27]. Fish collagen was identified as an allergen during the early 2000s, which may be a limiting factor for consumption of fish derived gelatine in

sensitive populations. The T-cell epitopes present in collagen are likely to be resistant to digestion by proteolytic enzymes, potentially inducing sensitization [28].

## 2.2. Comparison of Fish and Mammalian Gelatine

Physical and chemical properties of mammalian gelatines have been extensively researched and although fish-derived gelatines have also been extensively studied, the majority of results have been published recently [17,21]. Fish derived gelatine is a viable alternative to mammalian sourced gelatine. There is however a number of limitations to widespread use of fish gelatine, and they are related to difference in physical and chemical properties in comparison to mammalian-sourced gelatines. Odour, colour, techno-functional and film forming properties, as well as consistency of amino acid composition may significantly vary depending on the gelatine source. Therefore, great care should be paid to optimization of the production process in order to obtain a product with the best properties for intended applications. Gelatine quality is rated based on numerous parameters, such as solubility, transparency, colour, odour and taste, and functional properties including rheology, moisture, ash, protein, pH, setting point and time, melting point and time, gel strength and viscosity. Physical and chemical properties of gelatine are mostly influenced by the animal species from which they are derived. It is known that, in general, fish-based gelatines have lower melting temperatures and strengths compared to their commercial pig skin and bovine counterparts [29]. Warm water fish gelatine is reported in the literature to have better functional properties than cold-water fish gelatines [30,31]. The principal reason for these differences is that, in general, fish gelatines have a lower content of imino-acids (hydroxyproline and proline) than mammalian gelatines. Therefore, gelatine with low levels of imino acids tends to have lower gel strengths and melting points. The molecular weight distribution is also important in determining the gelling behaviour of gelatine. Muyonga et al. [32] reported that the content of hydroxyproline and proline is approximately 30% in mammalian gelatines, 22–25% in warm water fish gelatines, and only around 17% in cold water fish gelatines (such as cod). Relative lack of these amino acids is partially compensated for by higher concentrations of serine and threonine. For this reason, gelatines obtained from cold water fish act as viscous liquids at room temperature, limiting their use in the food industry [30]. Higher amount of hydrophobic amino acids can, however, be a potential advantage in certain scenarios. Avena-Bustillos et al. [30] have investigated water vapour permeability of cold- and warm-water fish skin gelatines films and compared them with different types of mammalian gelatines. Films obtained from cold-water fish species (*Alaskan Pollock* and salmon) gelatines showed lower water vapour permeability compared to warm water fish and mammalian gelatines. The authors concluded that, although physical properties of these gels were inferior, the lower water vapour permeability of fish gelatine films can be useful particularly for applications related to reducing water loss from encapsulated drugs and refrigerated or frozen foods. However, a contrasting report has been published by Atma [31] on the comparison of amino acid and proximate composition in several warm water fish species. Among the investigated fish species, King weakfish and Lizard fish were found to have the highest hydroxyproline and protein content, which did not correspond to their respective gel strengths. The author has concluded that imino acid content may not be the main factor influencing gel strength in all cases, and that multiple other factors, including other amino acids, extraction conditions and molecular weight distribution may also play an important role in gelatine production.

The most widespread application of gelatine in the food industry is the use in formulation of water-based gel desserts, owing to its unique property of melting at body temperature [7,21]. Fish-based gelatines have a disadvantage in this regard due to their lower gel strength and melting temperature. For this reason, numerous attempts have been made to improve their gel-forming and viscoelastic properties. This can be overcome by increasing gelatine concentrations or by using gelatine mixtures (of cold and warm-water fish). Zhou & Regenstein [33] have compared different textural properties of gelatine desserts obtained from cold- (*Alaskan pollock*) and warm-water (tilapia) fish species with commercial mammalian-based gelatines. Gel strength and rheological properties

of cold-water fish gelatines were less desirable compared to pure pig skin and tilapia gelatines, but mixtures of said gelatines exhibited much improved properties. The authors concluded that desserts made from fish gelatines would be more similar to desserts made from high bloom pork skin gelatine by (a) increasing the concentration of gelatine or (b) by using gelatine mixtures. In addition, the gel desserts made from fish gelatines melted at lower temperature, which may accelerate the flavor release in such food products. Although cold-water fish gelatines tend to possess lower gel strength compared to warm-water fish gelatines, cold maturation time should also be considered when creating gelatine-based products. Gómez-Guillén et al. [29] have reported on the importance of prolonged maturation at low temperature in the case of hake gelatine. They concluded that longer maturation time might be required to allow growth of existing nucleation sites within gelatine, since cold-water fish gelatine possesses a lower percentage of β- and γ- components compared to individual α-chains as found in hake gelatine.

The gelling temperature of cold-water fish gelatine is usually below 8–10 °C, which enables it to be used as a base for light-sensitive coatings, since it is a good medium for precipitation of silver halide emulsions at lower temperature than warm-blooded animal gelatine [34]. On the other hand, this limits the use of such gelatines as gelling components in food production. Despite being a techno-functional disadvantage, lower melting and setting points of fish gelatine may be useful in development of certain food products, due to a better release of aromas and imparting stronger flavour [35]. Absorption of ingested fish collagen is up to 1.5 times more efficient, indicating its superior bioavailability over bovine or porcine types. Due to its more efficient absorption, it is considered to be the best source of collagen for pharmaceutical applications [36].

Religious concerns and disease outbreaks including bovine spongiform encephalopathy (BSE) have resulted in a desire for gelatine replacement hydrocolloids and alternatives to mammalian sourced gelatine. Although the physical properties of most of the cold-water fish sourced gelatines are not ideal compared to mammalian gelatines (pig skin, cattle hide) their advantage is almost universal acceptability in terms of religious beliefs [37]. Eleven of the major gelatine and collagen peptide manufacturers in Europe are part of The Gelatine Manufacturers of Europe (GME), an association founded in 1974. More than 98% of the European gelatine and collagen production is accounted by these manufacturers, which amounts to 33% of the total worldwide production [23].

## 3. Production Strategies for Gelatine

Industrial production of gelatine is a well-known process, and in general, includes raw material washing, pre-treatment, extraction, and purification followed by drying and packing of the final product. Although the parameters of the steps vary greatly between manufacturers, the choice of raw material dictates the pre-treatment procedure and influences the complexity of production. Unlike bovine and porcine sources, fish skins used for industrial production of gelatine are often not subjected to harsh pre-treatment, due to weaker bonds in this type of collagen. A simplified scheme of fish gelatine production is shown in Scheme 1 [38].

```
┌─────────────────────────────────┐
│       Raw material washing      │
└─────────────────────────────────┘
                 ↓
┌─────────────────────────────────┐
│    Addition of water and acetic │
│      acid (pH set to 4.5-5.5)   │
└─────────────────────────────────┘
                 ↓
┌─────────────────────────────────┐
│    Extraction at 88-93 °C for 3-6│
│    hours (2 extraction cycles)  │
└─────────────────────────────────┘
                 ↓
┌─────────────────────────────────┐
│   Extract clarification (filtration│
│   through diatomaceous earth)   │
└─────────────────────────────────┘
                 ↓
┌─────────────────────────────────┐
│    Anion exchange treatment to  │
│       remove soluble salts      │
└─────────────────────────────────┘
                 ↓
┌─────────────────────────────────┐
│     Liquor concentration (to 44-│
│        46% solids; w/w)         │
└─────────────────────────────────┘
                 ↓
┌─────────────────────────────────┐
│     Product drying (by infrared │
│    heating), grinding and packing│
└─────────────────────────────────┘
```

**Scheme 1.** Basic steps of the fish gelatine production process [38].

To properly assess the economic feasibility of industrial-scale fish gelatine production, numerous factors, such as raw material availability and price, production costs, and final product price margin, need to be accounted. Although fish gelatine amounts to only a fraction of worldwide gelatine manufacturing, the high quantities of by-products generated by fisheries represent a potentially lucrative opportunity for its market increase. Recent work of the Trash2Cash project (2011–2015) in Denmark has undertaken considerable research concerning the economic feasibility of gelatine production from fish sources [39,40]. Findings from this project show that the market for fish gelatine and fish collagen hydrolysates is small, (2000 to 3000 tons per year), and that prices of final products vary from 10 to 15 € per kg, depending on traceability, degree of hydrolysis, taste, and purity [39]. As a part of the same project, financial and economic aspects of construction of a fish gelatine plant have been evaluated. Using a "greenfield" model (model which assesses costs of constructing a plant from

nothing at starting point—i.e., "green field") estimates of investments, operating costs and revenues were made [40]. In general, the estimation showed that, when major equipment and variable costs are taken into account, the final revenue would operate with a financial margin of almost 50%, provided that the operation of the plant is at full capacity. This operating revenue is estimated with fish gelatine prices set between 10–12 €/kg, in the case that the raw material (fish skin) costs are 2.25–2.5 DKK (0.30–0.34 €) per kilogram and yield of produced gelatine is 10% [40]. These estimations indicate that market prices of raw material and produced gelatine have the most pronounced influence on the final operating revenue. However, the expected yield of gelatine extraction can also be a major factor for considerations since it is dependent on multiple variables, such as raw material quality, composition, and origin. Having this in mind, careful optimization of production steps (pre-treatment, extraction) has to be taken into account for future production planning.

During gelatine production, the insoluble native collagen must be pre-treated before it can be converted into a form suitable for extraction [7,21]. This is routinely done by heating in water at temperatures higher than 45 °C. A chemical pre-treatment is intended to break non-covalent bonds in order to disorganize the protein structure, and produce adequate swelling and collagen solubilisation [7,17]. The production process variables (pH, temperature and time) during pre-treatment and extraction steps also have a significant influence on the collagen denaturation, which, along with the animal species and tissue type, affects the properties of the obtained gelatine [21,29].

### 3.1. Pre-Treatment and Extraction Strategies

Differences in the available literature are seen between different pre-treatment procedures regarding the same type of fish material (skin, bones, offal). In general, during the production of gelatine, the pre-treatment steps are important for weakening the chemical bonds between collagen chains and make it more suitable for subsequent extraction. There are two main pre-treatments used in the gelatine industry today: (a) Acid pre-treatment, which is done by treatment of the material with diluted acids. It is suitable for materials with less cross-linked collagen, like pig skin, and results in the so called type A gelatine (with isoelectric point at pH 6–9) [41]. Acid pre-treatment is also necessary in the case of gelatine production from bones, where it ensures the removal of bone mineral components prior to extraction; (b) Alkali pre-treatment, which is achieved by soaking of the treated material with diluted alkali solutions (NaOH, KOH, $Ca(OH)_2$.). It is commonly used as a pre-treatment of materials with highly cross-linked collagen, such as bovine hides. Gelatine obtained by this type of pre-treatment is called type B, with an isoelectric point at pH 5 [41]. Various types of pre-treatment and extraction strategies for gelatine isolation from marine/freshwater sources are shown in Table 2.

**Table 2.** Examples of gelatine pre-treatment and extraction strategies.

| Authors/Year | Material | Pre-Treatment | Extraction |
|---|---|---|---|
| [42] | African catfish (*Clarias gariepinus*) skin | NaOH at various concentration and time range (0.15–0.35% (w/v) and 40–120 min); Sulphuric acid at various concentration and time range (0.08–0.35% (w/v) and 40–120 min); Citric acid at various concentration and time range (0.6–1.4% (w/v) and 40–120 min) | Water at various temperature and time range (33–67 °C and 4–14 h) |
| [43] | Swim bladders of catla (*Catla catla*) | 0.15% NaOH (w/v) for 40 min; sulphuric acid (0.15%, v/v) and citric acid (0.5%, v/v) for 40 min (×2) | Water, 45–50 °C for 17 h |
| [44] | Dover sole (*Solea vulgaris*) skin | (a)  Acetic acid 0.05 M <br> (b)  Lactic acid at various concentrations (0.01, 0.025, 0.05 M) | Water, 45 °C overnight |
| [45] | Tuna (*Thunnus thynnus*) head bones | Alkaline protease from *Bacillus mojavensis*, 50 °C for 4 h; 0.4 M HCl for 7.5 h; 0.9% Ca(OH)$_2$ (w/v) for 144 h | Water, 75 °C for 4 h |
| [4] | Skins of several marine species (kerapu (*Epinephelus sexfasciatus*), jenahak (*Lutjianus argentimaculatus*), kembung (*Rastrelliger kanagurta*), kerisi (*Pristipomodes typus*) | 0.2% NaOH (w/v) for 40 min; sulphuric acid (0.2%, v/v) and citric acid (1%, v/v) for 40 min (×2) | Water, 45 °C for 18 h |
| [41] | Brownstripe red snapper (*Lutjanus vitta*) and bigeye snapper (*Priacanthus macracanthus*) skin | 0.2 M NaOH (3 × 30 min); 0.05 M acetic acid for 3 h | Water, 45 °C for 12 h |
| [46] | Mackerel (*Scomber scombrus*) and blue whiting (*Micromesistius poutassou*) bones | (a)  0.1 N NaOH for 30 min; 0.25 M HCl for 18 h <br> (b)  Flavourzyme/alcalase at an enzyme/substrate ratio of 0.1% (v/w) for 4 h (50 °C); 0.25 M HCl for 18 h | Water, 45 °C for 18 h |
| [47] | Clown featherback (*Chitala ornata*) skin | 0.1 M NaOH for 2 h; 0.05 M acetic acid for 30 min | Water at various temperature and time range (45, 65, 85 °C and 6 h and 12 h) |
| [48] | Baltic cod (*Gadus morhua*) skin | No pre-treatment (only manual cleaning of material) | Water at various temperature and time range (30–60 °C and 15–120 min) |
| [32] | Nile perch (*Lates niloticus*) skin and bone | Skin: 0.01 M sulphuric acid (pH of 2.5–3.0) for 16 h Bones: 3% HCl for 9–12 days | Three sequential extractions for 5 h, at 50, 60 and 70 °C; followed by boiling for 5 h |
| [20] | Splendid squid (*Loligo formosana*) skin | 0.05 M NaOH for 6 h; 0.05 M phosphoric acid for 24 h | Water, with different temperatures (50, 60, 70 and 80 °C) |

**Table 2.** *Cont.*

| Authors/Year | Material | Pre-Treatment | Extraction |
|---|---|---|---|
| [49] | Tilapia (*Oreochromis niloticus*) skin | 0.3 M NaOH for 1 h; HCl, citric and acetic acid at various concentrations (0.01–0.20 M) | Water, 50 °C for 3 h |
| [19] | Herring species (*Tenualosa ilisha*) skin | 0.2 M Ca(OH)$_2$ for 1 h; 0.1 M citric acid for 3 h | Water, 50 °C for 3 h |
| [50] | Ribbon fish (*Lepturacanthus savel*) surimi processing waste | 0.2 M Ca(OH)$_2$ for 1 h; 0.1 M citric acid containing bromelain in various concentrations for varying times | Water, at different combinations of temperatures and durations |
| [51] | Red snapper (*Lutjanus campechanus*) and grouper (*Epinephelus chlorostigma*) bones | 0.2% NaOH (w/v) for 45 min; sulphuric acid (0.2%, v/v) and citric acid (1%, v/v) for 45 min (×2) | Water, 45 °C for 24 h |
| [35] | Skins of dog shark (*Scoliodon sorrakowah*), skipjack tuna (*Katsuwonus pelamis*) and rohu (*Labeo rohita*) | 0.1 M NaOH for 2 h; 0.05 M acetic acid for 24 h | Water, 45 °C for 12 h |
| [52] | Seabass (*Lates calcarifer*) skin | 0.1 M NaOH for 3 h; 0.05 M acetic acid for 2 h | Water at various temperature and time range (45, 55 °C and 3, 6 and 12 h) |
| [53] | Alaska Pollock skin | NaOH/Ca(OH)$_2$ at various concentrations for 60 min; acetic, citric and sulfuric acid at various concentrations for 60 min | Water, 50 °C for 3 h |

### 3.1.1. Chemical Pre-Treatment

The extraction conditions, such as temperature, time and the chemicals used, can influence the functional properties of gelatine, by producing varying lengths of polypeptide chains [48]. The degree of collagen cross-linking in the raw material is a principal factor in the choice of the pre-treatment process during gelatine manufacture, and is highly dependent on a number of factors, such as collagen type, tissue, animal species, and age [54]. In the case of fish skins, acid pre-treatment may be considered as sufficient, and numerous authors have used it as the only form of pre-treatment. Gómez-Guillén et al. [7] have investigated chemical and physical properties of gelatine obtained from several different marine species, under mild swelling conditions using 0.05 M acetic acid as pre-treatment, followed by extraction in distilled water at 45 °C overnight. Their results showed that gelatines from flat-fish species (sole and megrim) possessed higher strength and thermostability than those obtained from cold-water fish species (cod and hake). Lactic acid at a concentration of 0.025 M has been found to be suitable for pre-treatment of fish skins instead of the commonly used acetic acid [54]. Higher concentrations of lactic acid (0.05 M), however, increase the level of hydrolysis and therefore adversely affected the gel strength and viscoelastic properties. Citric acid may also be used for the manufacture of food-grade gelatine from fish skin since it does not impart undesirable sensory properties (colour, odour) to the extracted gelatine. Gómez-Guillœn and Montero [55] have investigated the influence of several organic acids on the properties of gelatine extracted from megrim (*Lepidorhombus boscii*) skin. They concluded that, among all tested organic acids, acetic and propionic acid extracts produced gelatine with the best properties including viscoelastic, setting and melting temperatures, and gel strength properties. Although pre-treatment with citric acid has been shown to produce the least turbid gelatine, its physical properties were inferior to other investigated acids. The influence of different acid pre-treatments was also investigated by Niu et al. [49] on gelatine obtained from tilapia (*Oreochromis niloticus*) skin. The authors concluded that the concentration of used acid had significant influence on gelatine recovery, gelatine viscosity, and molecular weight distribution. Gelatine prepared using too low or too high a concentration (e.g., 0.01 M or >0.05 M HCl or citric acid) yielded a product with a lower ratio of large molecule components, such as β-chains, and exhibited lower viscosity.

In the case when fish skin is used as a material for gelatine extraction, it is known that combinations of alkali and acid pre-treatments have positive effects on the final product properties, and this type of pre-treatment has been patented by Grossman et al. [56]. Zhou and Regenstein [53] have shown that combinations of acid and alkali pre-treatment had a positive impact on the yield and gel strength of gelatine extracted from Alaska Pollock. Shyni et al. [35] have reported on physical and chemical differences between gelatines extracted from skins of dog shark (*Scoliodon sorrakowah*), skipjack tuna (*Katsuwonus pelamis*), and rohu (*Labeo rohita*). Their results show that dog shark skin gelatine had the most optimal yield and gel strength, as well as other physical and chemical properties (molecular weight, viscosity, melting point, foaming properties, water holding capacity, odour, colour and clarity) compared to tuna and rohu gelatine, which could be explained by its high content of hydroxyproline. Alkali pre-treatment is useful for removal of non-collagen proteins and fats, while subsequent treatment with diluted acids provides mildly acidic pH of the medium which enables a good yield of gelatine extraction [35,57]. Gómez-Guillén et al. [58] have reported that application of high pressure (250 and 400 MPa) either during acid pre-treatment or during water extraction enabled significant shortening of the duration of time required for those steps, obtaining good yield of gelatine in only a few minutes. Other collagen-rich tissues in fish by-products may also be a feasible source of gelatine, especially if their industrial output is sufficiently abundant. Extraction of gelatine from swim bladders of catla (*Catla catla*) using mild pre-treatment with NaOH, sulphuric, and citric acid is reported by Chandra and Shamasundar [43]. The obtained gelatine in their study had a satisfactory yield (13.5% (w/w)) and good gel strength (264.6 g), indicating that fish swim bladders can also represent an underused source for production of fish gelatine.

Besides from fish skin, gelatine can also be extracted from mineralized structures such as fins, scales, and bones. Although fish bone and scale represent a valuable source of gelatine, additional demineralization should be introduced prior to gelatine extraction due to the high mineral content of these tissues. Diluted hydrochloric acid is most often used for bone demineralization [45,46,51], although other compounds, such as EDTA, have also been used for this purpose [59,60]. Although recoveries of gelatine extracted from bones and scales are usually lower in comparison to skin gelatines of the same species, bones and scales are nevertheless an important sources due to their high percentage in the total industrial output of fish by-product generated from surimi production [7]. Therefore, care must be taken in order to optimize the pre-treatment methods for such composite samples in order to obtain the highest yield of gelatine with the best properties.

### 3.1.2. Enzymatic Pre-Treatment

Treatment with proteolytic enzymes, either alone or in combination with other pre-treatments (alkaline, acidic, etc.) is another option for improving extraction yield and quality of the obtained product. Enzymes are catalyst biomolecules which can speed the rate of biological reactions by catalyzing a transition state with a lower energy of activation. They can also hydrolyze the covalent cross-links in the terminal regions of proteins and faciliate the transformation of collagen to gelatine, while producing less waste compared to the chemical treatments [61]. Khiari et al. [46] have compared properties of gelatine extracted from bones of mackerel and blue whiting obtained using non-enzymatic (HCl) and enzymatic pre-treatment using Flavourzyme (fungal protease/peptidase complex obtained from *Aspergillus oryzae*). They concluded that gelatine obtained by enzymatic pre-treatment of bones showed significantly higher emulsifying activity (EAI) and stability (ESI) indices in comparison to acid pre-treatment. Gelatin extraction from bigeye snapper (*Priacanthus tayenus*) skin was developed by Nalinanon et al. [62], using a pepsin-aided process (big eye snapper pepsin, BSP) in combination with a protease inhibitor (pepstatin A and soybean trypsin inhibitor). The bloom strength of pepsin-treated gelatine was greater than the gelatine extracted from bigeye snapper skin by the conventional process, which had a substantial degradation of gelatine components, and soybean trypsin inhibitor added during the extraction process significantly reduced the degradation of α- and β-chains in the gelatine. Since most proteolytic enzymes are usually able to cause significant degradation of gelatine α- and β-chains, careful optimization of pre-treatment conditions is required to avoid this. Zhang et al. [61] have investigated pre-treatment optimization of grass carp fish (*Ctenopharyngodon idella*) scales by protease A2G enzyme utilizing the response su rface methodology (RSM). The resulting gel strength (276 ± 12 g) and viscoelastic properties were comparable to porcine skin gelatine at lower temperatures, while the imino acid content, gelling and melting points were lower. Since surimi processing wastes represent composite material of skin, scale, bone and muscle, enzymatic pre-treatment may be a good solution for removal of non-collagenous proteins prior to gelatine extraction. Enzymatic digestion can also be used as part of the pre-treatment, to remove interfering tissues before a more conventional chemical treatment is used. Haddar et al. [45] have used alkaline protease from Bacillus mojavensis in their work on extracting gelatine from tuna (*Thunnus thynnus*) heads, where the enzyme was used to obtain clean bone material before demineralisation with HCl and subsequent treatment with $Ca(OH)_2$.

### 3.1.3. Extraction of Gelatine

After pre-treatment of fish skins, extraction of gelatine with water at various temperatures and time lengths is the universally applied approach for obtaining gelatine. Karim and Bhat [17] and Karayannakidis and Zotos [21] have reported on the various procedures employed for gelatine pre-treatment and extraction. Most commonly, distilled water was used and the temperatures and lengths of extraction show a high variability between different authors. The most often used extraction temperature in various research papers is around 45 °C, with the time of the extraction varying from 12 to 18 h (or "overnight") [43,48,51,54]. Multi-stage extractions and different temperatures have also been reported [32,63–65]. Hou and Regenstein [63] have developed an optimized method for pre-treatment and extraction of gelatine from Pollock skin. They concluded that an extraction

temperature of 50 °C was optimal regarding the extraction yield. Besides from pure water, some authors have reported successful gelatine extraction using mild acidic conditions [66] and also with addition of mixtures of protease inhibitors [53]. Due to the low denaturation temperature of fish collagen, the extraction temperature and time can have a significant influence on the properties of the extracted gelatine, especially on the gel strength. Gel properties of gelatine from clown featherback skin under different extraction temperatures (45, 65 and 85 °C) and times (6 and 12 h) were investigated by Kittiphattanabawon et al. [47]. Their results indicated that, although yield was highest at the highest extraction temperatures, by increasing temperature and prolonging extraction time, band intensity of α-, β- and γ-chains decreased in the extracted gelatines. Similar findings were reported by Alfaro et al. [42], where temperature, extraction time, and concentration of acid during pre-treatment were used to assemble a central composite rotational design (CCRD) in order to elucidate its effect on gelatine viscosity. The strong influence of pre-treatment and extraction conditions on the yield and properties of fish gelatine need to be taken into consideration in an industrial setting, and usually a compromise between yield, desired properties, and energy efficiency needs to be considered for optimal production.

### 3.2. Improving the Properties of Fish Gelatine

Although there has been an increasing demand for fish gelatine due to its religious and safety advantages over pig and bovine sources of gelatine, the main limiting factors of its widespread use lies in its technofunctional properties—i.e., the lower gel strength and melting temperatures compared to those for mammalian gelatines. This poses a challenge for commercial exploitation, and various approaches have been proposed to date to overcome these issues. Ultraviolet (UV) irradiation represents a physical, cost-effective, non-thermal, and environmentally friendly technology that has received increased attention in the food sector during recent years. Bhat and Karim [67] have investigated the effect of UV irradiation (at 30 and 60 min interval lengths) on the gel strength of fish gelatine granules. They concluded that the irradiated samples exhibited significant improvements in the gel-strength, a reduction in viscosity, as well as changes in the melting enthalpy. These results indicate the possibility of using simple UV radiation as a method to improve cold fish gelatine properties. In their more recent work, Bhat and Karim [68] have also investigated combination of UV irradiation and addition of sugars (ribose and lactose) on the properties of fish gelatine based films. Their results indicated that films with added ribose showed decreased solubility after UV treatment and exhibited higher swelling percentages than films with added lactose. Otoni et al. [69] have also noted an improvement in functional properties of of fish gelatines from cold- (cod, haddock, pollock) and warm-water (tilapia) fish as a consequence of UVB radiation exposure.

Gelling properties of fish based gelatines may further be modified by use of various chemical agents which induce molecular crosslinking, such as glutaraldehyde [70], as well as by creating mixtures with various non-gelatine systems such as pectin [71]. Besides from natural polymers, several synthetic polymers have been used to create gelatine hybrid hydrogels. Zohuriaan-Mehr et al. [72] have reported a number of organic (PEG-dialdehyde, acrylamines, EDTAD, poly(acrylic acid)), and inorganic (kaolin, silica gel) compounds which can affect gel strength, solubility, and hydrophobicity of such composite hydrogels. Another means of improving gelling properties of fish gelatine is to introduce enzymatic crosslinking using transglutaminase. This enzyme catalyses the formation of crosslinking bonds between γ-amide groups of glutamine and ε-amino groups of lysine. Baltic cod gelatine treated with transglutaminase was shown to be able to withstand heating in boiling water for 30 min without melting [48]. As a collagen denaturation product, gelatine contains many divalent metal ions such as calcium, copper, iron, and zinc. These ions can form ionic bonds with the gelatine carboxylic acid groups, thus influencing the organization of the gelatine network. Removal of those metal ions by means of ion-exchange may improve further crosslinking between gelatine molecules, as demonstrated by Xing et al. [73] who purified gelatine solutions using Chelex resin to replace divalent metal ions with sodium ions prior to crosslinking by 1-ethyl-3-(3-dimethylaminopropyl) carbodiimide (EDC). On the other hand, the effect of different salts on the rigidity or melting temperature of animal gelatines has

also been researched previously [74,75]. Koli et al. [75] have optimized a method for improving fish gelatine extracted from Tiger-toothed croaker (*Otolithes ruber*), using combination of three co-enhancers (MgSO$_4$, sucrose, and transglutaminase). By addition of co-enhancers at optimal concentrations in their experiments, the gel strength and melting point were improved from 170 to 240.89 g and 20.3 to 22.7 °C, respectively. Due to their better acceptability by consumers, natural compounds and extracts can also be used to improve gelatine properties. Araghi et al. [76] examined the effects of natural phenolic cross-linkers (ferulic and caffeic acid) on fish gelatines. In their study, caffeic acid had notable effects in decreasing solubility, water vapour permeability, and oxygen permeability of fish gelatine films. Natural phenolic compounds may therefore be used as a natural ingredient for increasing safety of gelatine-based biodegradable packaging, by improving their barrier and physicochemical properties. Another natural material, chitosan nanoparticles (CSNPs), with excellent physicochemical properties, is known to be environmentally friendly, and bioactive, has been researched for improving properties of fish gelatine based films. Hosseini et al. [77] have created novel bio-nanocomposite films by addition of CSNP particles (created by ionic gelation between chitosan and sodium tripolyphosphate) into fish gelatine film matrix. Newly created films had significantly increased tensile strength and elastic modulus, and decreased water vapour permeability compared to fish gelatine films.

With the exception of its inferior physical properties when compared to mammalian counterparts, fish derived gelatine intended for food use often possesses undesirable sensory properties characterized by an unpleasant "fishy" flavour [78]. Sae-leaw, Benjakul, and O'Brien [79] have investigated the effects of defatting and tannic acid incorporation during extraction on the properties and fishy odour of gelatine obtained from seabass skin. They concluded that defatting by pre-treatment with citric acid and isopropanol and subsequent incorporation of tannic acid during the extraction prevented lipid oxidation and the subsequent development of volatile compounds and fishy odours in the resulting gelatine. The intensity of fishy odour may also increase if the storage of frozen raw materials is prolonged before processing, due to formation of volatile aldehydes and alcohols [78]. Therefore, delays in processing should be avoided in order to minimize formation of undesirable odour and further loss of technofunctioal properties of gelatine.

## 4. Opportunities for Novel Applications of Fish Gelatine and Collagen

Although gelatine has many applications in various industries, advances in food science, medicine and material science have yielded a number of novel applications. Due to its versatile physicochemical properties, high degree of biocompatibility and relatively low price, gelatine is an ideal material for numerous applications.

Tissue engineering has been an emerging field of modern regenerative medicine. Collagen has historically been utilized in biomedical field in treatment of tissue injuries, due to its property to act as a hemostatic agent. After discovery of its regenerative properties, it was applied in 3D cultures for use in regenerative medicine [80]. There has been a rapid development of scaffolds consisting of natural (collagen, gelatine), bioabsorbable syntethic (polyglycolic acid, polylactic acid) and inorganic (hydroxyapatite) polymers during recent years. [81]. In particular, collagen is the most promising material for tissue engineering due to its biocompatibility and biodegradability. However, due to the low denaturation and melting temperatures, collagen of most fish species is not suitable for such applications in its native form. For this reason, cross-linking of collagen by chemical or physical means is often studied for biomedical applications. Chemical treatments induce high strength and stability to the collagen matrix but they can result in potential cytotoxicity or poor biocompatibility, whereas physical treatments, such as UV irradiation may produce good stability and no cytotoxicity [81]. Nagai et al. [82] have prepared elastic vascular grafts from salmon collagen using mixtures of acidic collagen solution and fibrillogenesis-inducing buffer containing a cross-linking agent (water-soluble carbodiimide, WSC). These grafts induced little inflammatory reactions after subcutaneous placement in rat tissues. Collagen was also used as a matrix for research investigating the possibility of regeneration of dental pulp after pulpectomy, using stem cells [83]. Furthermore, 3D printing processes have found numerous applications, including biomedical. Fish gelatine, which is

more soluble and remains liquid at lower temperatures compared to mammalian gelatines is an good potential candidate for use a biological dye for use in 3D printing of tissue scaffolds [84]. Visser et al. [85] have created reinforced gelatine metha-acrylamide (GelMA) hydrogels with poly(e-caprolactone) (PCL) fiber scaffolds using melt electrospinning direct writing as a form of 3D printing. The stiffness and elasticity of the created structures have approached those of articular cartilage tissue.

Beside the use of gelatine in its native form, fish gelatine hydrolysates, obtained by enzymatic hydrolysis, offer an interesting option for by-product utilization by the fish-processing industry. Numerous companies worldwide offer fish gelatine/collagen hydrolysates for use in nutraceutical and for cosmetic purposes. Nutraceuticals are broadly defined as "food or part of a food that provides health benefits in addition to its nutritional content" [86]. They are pharmacologically active substances that can be obtained from food (of animal or vegetable origin) and can be further concentrated and used in a suitable pharmaceutical form [87]. Since an official and specific definition of nutraceuticals is still missing from legislative standpoint, many of the health and safety claims for these products are experiencing legal challenges worldwide [88]. Although the EU Commission has yet to approve many of the health and cosmetic claims, some manufacturers are already selling their products with certain claims supported by current research. Considering the higher cost of fish-derived gelatine in comparison to mammalian sources, production of bioactive products for specialized food and pharmaceutical use may represent a good opportunity for increasing its economic value. Such hydrolysates, consisting of various peptides, are relatively cheap and easy to produce, and many have shown to possess proven health and functional (antioxidant, antihypertensive, immunomodulatory and antimicrobial) benefits. Bioactive peptides from food proteins offer great potential for incorporation into functional foods and nutraceuticals [15,89]. Some of these products, such as sardine muscle hydrolysate, have already been approved by FDA and EFSA for use in human nutrition [15]. Lee et al. [90] have investigated angiotensin I converting enzyme (ACE I) inhibitory properties of tuna frame hydrolysates obtained by several proteolytic enzymes (alcalase, neutrase, pepsin, papain, α-chymotrypsin and trypsin). Their results showed that peptic hydrolysate exhibited the highest ACE-I inhibitory activity, and a potent ACE-I inhibitory peptide composed of 21 amino acids was subsequently isolated. Antioxidant activity of a hydrolysate from Nile tilapia (*Oreochromis niloticus*) skin gelatine was examined by Choonpicharn et al. [91]. Hydrolysates obtained by several enzymes (bromelain, papain, trypsin, flavourzyme, alcalase and neutrase) showed varying levels of antioxidant (ABTS radical scavenging, reducing power, ferrous ion chelating activity, inhibition of linoleic acid oxidation) activity and also a significant degree of ACE-I inhibitory activity. Beside their health benefits, fish gelatine hydrolysates also exhibit many useful techno-functional properties which may be utilized by the food industry. Hydrolysate of shark skin gelatine was tested as a cryoprotectant on surimi subjected to different freeze-thaw cycles by Kittiphattanabawon et al. [92], and the results indicated that gelatine hydrolysates with 10% degree hydrolysis was able to prevent the denaturation of surimi protein compared to a commercial cryoprotectant. Nikoo et al. [93] reported that a tetrapeptide isolated from Amur sturgeon skin gelatine showed antioxidative and cryoprotective effects in Japanese sea bass mince subjected to repeated freeze-thawing cycles. Such properties of gelatine hydrolysates have excellent potential for use by the food industry for improving shelf-life and oxidative stability of food products and commodities. Antimicrobial activity of fish gelatine hydrolysates has also been demonstrated by Hong et al. [94]. Alcalase-derived glycosylated hydrolysates of fish gelatine had antioxidative and antimicrobial activity when incubated with Escherichia coli and Bacillus subtilis, indicating its potential for use as anantimicrobial agent.

## 5. Conclusions

By-products from various marine processing industries represent an economic and environmental challenge, and solid processing leftovers are currently utilized for production of various low value products (animal feeds, fish oil, fertilizer, etc.) [6]. Gelatine is used in the food, pharmaceutical, and photography industry for a number of applications including jelly production, encapsulation,

and fruit juice clarification, dairy processing, soup manufacture, photography and others. Typical applications of gelatine depend on the gelatine type, and its great versatility enables use in both the food and pharmaceutical industry. Edible gelatine must meet the requirements laid by the Food Hygiene Regulation (EC) No 853/2004 (also Commission Regulation (EU) 2016/355 of 11 March 2016 amending Annex III to Regulation (EC) No 853/2004) and is additionally subject to European food regulations [23].

Production of gelatine from fishery by-products requires careful selection and optimization of pre-treatment and extraction steps in order to obtain optimum yield and physico-chemical properties. Numerous chemical, physical and enzymatic pre-treatment steps have been reported in the scientific literature, although current industrial scale production usually resorts to most cost-effective simple procedures. Depending on the intended use, properties of the fish derived gelatine may be further improved and modified using various chemical and physical processes which can impact its physical properties, such as bloom strength, elasticity and solubility. Beyond its well-established uses in food and pharmaceutical industry, fish gelatine has a potential use in several emerging fields, such as biomedical science (tissue engineering/3D printing), owing to its unique properties, good biocompatibility, and relatively low price. Beside the use of gelatine in its native form, fish gelatine hydrolysates, obtained by enzymatic hydrolysis, offer an interesting option for by-product utilization by the fish-processing industry. Such hydrolysates, consisting of various peptides, are relatively cheap and easy to produce, and many have been shown to possess proven health and functional (antioxidant, antihypertensive, immunomodulatory and antimicrobial) benefits. Numerous companies worldwide offer fish gelatine/collagen hydrolysates for use in nutraceutical and for cosmetic purposes, although the EU Commission has yet to approve many of the health and cosmetic claims. Based on the recent scientific advances in production and novel fields of potential use, gelatine derived from marine products represents an interesting option for industrial processors for adding economic value to fishery by-products in the future.

**Author Contributions:** Writing-Original Draft Preparation, I.M.; Writing-Review & Editing, M.H.; Project Administration, M.H.

**Funding:** This research was part of project "Fishbowl–production of clean label gelatin from boarfish" (sanction reference: DAFM/07/2017/PDFP) funded by Bord Iascaigh Mhara (BIM).

**Conflicts of Interest:** The authors declare no conflict of interest.

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
