# Peer review of "Marine Gelatine from Rest Raw Materials"

_applsci, doi:10.3390/app8122407_

Reviewer 1 Report

The manuscript entitled "Marine gelatine from rest raw materials" is intersting and fits the scope of the Journal. Is properly assessed being a review and adds information to the area of interest. One criticism ragards the use of the term "nutraceutical" which is undefined in the text. It would be useful to remark the context when using this term and possible perspective application. Please add the following references and add it to the reference list to better clear the term and use:

Santini A, Novellino E. Nutraceuticals: Beyond the Diet Before the Drugs. Current Bioactive Compounds, 2014, 10: 1-12.

Santini A, Tenore GC, Novellino E. Nutraceuticals: A paradigm of proactive medicine. European Journal of Pharmaceutical Sciences. 2017. 96: 53-61.

Santini A, Cammarata SM, Capone G, Ianaro A, Tenore GC, Pani L, Novellino E. Nutraceuticals: opening the debate for a regulatory framework. British Journal of Clinical Pharmacology. 2018. 84: 659-672.

Author Response

Dear Reviewer and editor,

We have taken on board the reviewers comments and we agree that the omission of a reference to nutraceuticals is important. In line with the request of the reviewer we have added the references and give an explanation about what a nutraceutical is in the revised manuscript. Please see revised manuscript where we cite the references listed by the reviewer.

Reviewer 2 Report

Review on manuscript:

Marine gelatine from rest raw materials by  Ivan Milovanovic and Maria Hayes submitted to Applied Sciences

In the manuscript submitted for comments the Authors reviewed the feasibility of producing gelatine sourced from marine processing by-products using different pre-treatment and extraction methods with the aim of improving the technological and functional properties of the final product and improving the clean-label status of gelatines. In my opinion the manuscript is prepared properly and after a minor revision can be published in the Applied Sciences journal.

Detailed recommendation:

lines 27-28 – sentence should be rewritten, exactly the same begins the abstract,

lines 140, 315, 320, 330, 331 and next – for Latin names Italic style should be used

table 2 – could be presented in more compact way, quoting the first author's name and the year is not necessary, just the number of the reference item,

lines 744 and 749 – journal’s names should be written with capital letters.

Author Response

We agree with reviewer two and we have edited the text accordingly.

We have deleted the lines suggested by reviewer 2

We have used italics where suggested by reviewer 2

We have amended the references in Table 2

We have written the journal names in capital letters as suggested by reviewer 2